# Spatially heterogeneous dynamics in a metallic glass forming liquid imaged by electron correlation microscopy

Pei Zhang[1], Jason J. Maldonis [1], Ze Liu[2], Jan Schroers[2] & Paul M. Voyles [1]

Supercooled liquids exhibit spatial heterogeneity in the dynamics of their fluctuating atomic arrangements. The length and time scales of the heterogeneous dynamics are central to the glass transition and influence nucleation and growth of crystals from the liquid. Here, we report direct experimental visualization of the spatially heterogeneous dynamics as a function of temperature in the supercooled liquid state of a Pt-based metallic glass, using electron correlation microscopy with sub-nanometer resolution. An experimental four-point space-time correlation function demonstrates a growing dynamic correlation length, $\xi$, upon cooling of the liquid toward the glass transition temperature. $\xi$ as a function of the relaxation time $\tau$ are in good agreement with Adam-Gibbs theory, inhomogeneous mode-coupling theory and random first-order transition theory of the glass transition. The same experiments demonstrate the existence of a nanometer thickness near-surface layer with order of magnitude shorter relaxation time than inside the bulk.

[1] Department of Materials Science and Engineering, University of Wisconsin-Madison, Madison, WI 53706, USA. [2] Department of Mechanical Engineering and Materials Science, Yale University, New Haven, CT 06511, USA. Correspondence and requests for materials should be addressed to P.M.V. (email: paul.voyles@wisc.edu)

The dynamics of the rearrangements of atoms or molecules in a liquid are challenging to determine directly, yet they govern critical processes including diffusion[1], viscous flow[2], nucleation and growth of crystal phases[3], and the glass transition[4,5]. In the supercooled state, it is envisioned that dynamics become spatially heterogeneous[6], with nanoscale domains exhibiting widely varying characteristic structural relaxation times[7–12]. According to most microscopic theories, the relaxation time and characteristic length scale of slow domains grows as the liquid cools through the glass transition[13–16]. In contrast, classical nucleation theories typically treat the liquid prior to the transition to a crystal as a uniform interface[17]. Experimental data on liquid dynamics comes from their frequency-dependent response to mechanical, electrical, or thermal stimuli[18–20], spectroscopy[21–23], or scattering of coherent light[24–27]. These are bulk measurements, from which spatial heterogeneity in dynamics largely inferred, with a few indirect exceptions[28].

Electron correlation microscopy (ECM) is a method for measuring liquid-state dynamics with nanoscale spatial resolution[29,30]. It uses time-resolved coherent electron scattering to study liquid dynamics at the nanoscale[29,30]. When performed in diffraction mode with a focused probe, it is the electron equivalent of (X-ray) photon correlation spectroscopy[23,31]. Coherent scattering produces a speckle pattern, $I(\mathbf{k}, t)$, where $I$ is the intensity, $\mathbf{k}$ is the scattering vector, and $t$ is time. Each speckle corresponds to a volume of the sample at the length scale of probe size with sufficient internal order to create constructive interference of the scattered waves. Local structural rearrangements cause intensity fluctuations of the speckle, so the lifetime of a speckle represents the time over which a particular structure persists. Statistical analysis of many speckles using the time autocorrelation function

$$g_2(t) = \frac{\langle I(t')I(t' + t)\rangle}{\langle I(t')\rangle^2},\qquad(1)$$

where $t'$ is the time of a frame in the diffraction time series, $t$ is delay time after $t'$, and $<>$ denotes average over all $t'$, fit to the Kohlrausch–Williams-Watt (KWW) equation

$$g_2(t) = 1 + A\exp\left[-2\left(\frac{t}{\tau}\right)^{\beta}\right],\qquad(2)$$

yields the characteristic relaxation time $\tau$ and the stretching exponent $\beta$[32]. The advantage of ECM is spatial resolution. A modern transmission electron microscope (TEM) can create sub-nanometer diameter beams with high coherence[29,33], and the electron's large scattering cross-section creates nanodiffraction patterns with acceptable signal to noise ratio even from small volumes.

Previous ECM experiments were performed using a stationary, nanometer diameter probe beam, so they measured $I(\mathbf{k}, \mathbf{r}, t)$ at only one point $\mathbf{r}$ on the sample. Figure 1a shows how ECM implemented using dark-field TEM imaging can measure many positions in parallel, producing an image of the liquid state dynamics for an entire sample. A broad coherent electron beam illuminates the sample, and a diffraction pattern $I(\mathbf{k}, t)$ is formed in the back focal plane of the objective lens. A small objective aperture is introduced, blocking most of the pattern, but admitting a speckle or two at a particular $\mathbf{k}$. Additional optics form a real-space image from just the scattered electrons in those speckles, producing a spatial map of the speckle intensity, $I(\mathbf{k}, \mathbf{r}, t)$. This represents the same basic data as probe-based ECM, but with one point in $\mathbf{k}$ instead of many, and many points in $\mathbf{r}$ instead of one (the difference is exactly analogous to the difference between probe-based and dark-field image based fluctuation electron microscopy[34]). A time series of images enables us to track the speckle intensity at a particular position on the sample, revealing the dynamics at that position.

Here, we apply ECM using dark-field TEM imaging to the supercooled liquid state of Pt-based metallic glass nanowires and visualize the relaxation dynamics at sub-nanometer spatial resolution. The ECM data provide direct evidence of spatially heterogeneous dynamics in the supercooled liquid state. A four-point correlation function derived from the data exhibits a

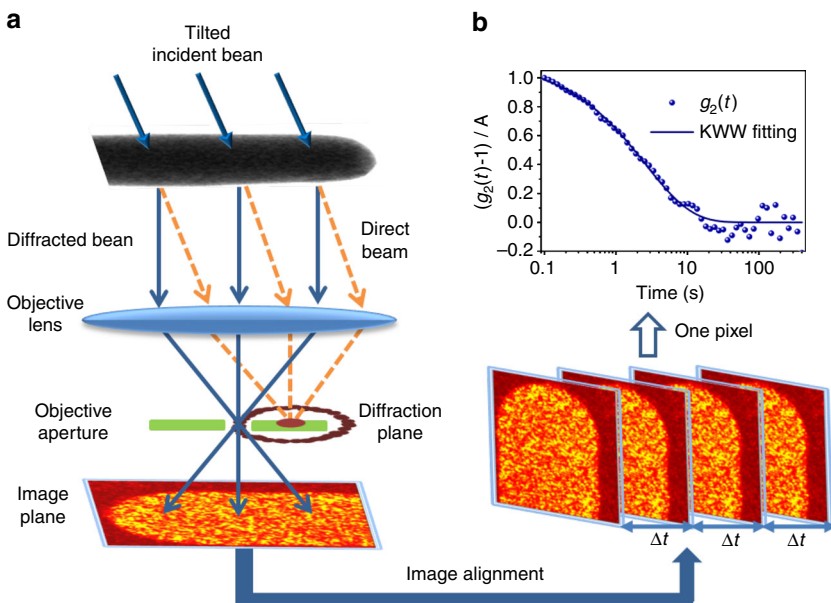

**Fig. 1** Tilted dark-field electron correlation microscopy. **a** Schematic of the experiment. Tilted illumination shifts the transmitted beam off the optic axis of a microscope. A small, on-axis objective aperture selects one speckle in the diffraction pattern, forming a real-space image. Structural rearrangements cause fluctuations in the intensity of the image speckles. A time series of ~4000 dark-field images is recorded and then aligned to correct drift of the sample. **b** The time autocorrelation function $g_2(t)$ is calculated from the intensity time series at every pixel, then fit to the KWW equation to extract the relaxation time $\tau$ and stretching parameter $\beta$. The figure shows a typical $g_2(t)$, which is well converged. This calculation is repeated for every pixel in the image

growing dynamical length scale as the liquid cools toward the glass transition temperature, consistent with theories of the glass transition[14,15,35]. We also identify a sub-nanometer thick near-surface layer with dynamics an order or magnitude faster than the bulk and show that it may influence crystallization of the wires[36]36.

## Results

**Dark-field ECM on metallic glass nanowires**. ECM experiments were performed on $Pt_{57.5}Cu_{14.7}Ni_{5.3}P_{22.5}$ nanowires[36], which are small enough to be used as TEM samples without requiring thinning with ion beams that can damage the sample surface, creating catalytic sites for crystallization. The wires were heated into the supercooled liquid state between the glass transition temperature $T_g = 507$ K up to 523 K, near the crystallization temperature[37]. Figure 1b shows an example of the normalized, resampled $g_2(t)$ calculated from the intensity fluctuations of one pixel inside a nanowire at 523 K. It shows good convergence to zero at long times and fits well to the KWW stretched exponential behavior.

**Spatial maps of structural relaxation time**. Fits like Fig. 1b to every pixel in the image yield spatial maps of $\tau$. Figure 2 shows representative images of $\tau$ as a function of temperature, calculated from $g_2(t)$ at every pixel. The images show striking visual evidence of nanoscale spatially heterogeneous dynamics in the supercooled liquid. At higher temperature, like Fig. 2a ($T_g + 16$

K/523 K), $\tau$ varies by an order of magnitude from place to place, with an apparently random distribution of domains ~1 nm in diameter. As the temperature approaches $T_g$, the $\tau$ distribution becomes wider, covering two orders of magnitude near $T_g$ (Fig. 2e, 507 K). The domains with similar $\tau$ also appear to grow in size, with more globular, extended domains several nanometers in diameter developing. At all temperatures, the histogram of relaxation times is well described by a log-normal distribution, characteristic of a random, non-negative process, the width of which increases with decreasing temperature (Supplementary Fig. 1 and Supplementary Note 1). The stretching exponents $\beta$ range from 0.2 to 1 and show no strong correlation with $\tau$ (Supplementary Figs. 1 and 2). $\beta < 1$ indicates a superposition of multiple relaxation processes with different characteristic times, which is typical for complex liquids near $T_g$ (e.g., ref. [23] for a metallic glass). With increasing temperature, rearrangements in liquids become less correlated and eventually only one relaxation process prevails, mathematically reflected by $\beta = 1$. The observed decrease of the mean $\beta$ with decreasing temperature approaching $T_g$ (Supplementary Fig. 1), indicates more dynamic heterogeneity, in line with the direct observation (Fig. 2). The temporal evolution of the domains is visualized using a sliding window along the time series (Supplementary Movie 3 and Supplementary Note 2). Spatial maps of relaxation time extracted from the first $20\tau_{med}$ and second $20\tau_{med}$ of a data set $40\tau_{med}$ in total length show different pattern but similar mean $\tau$ inside bulk (Supplementary Fig. 3 and Supplementary Note 3).

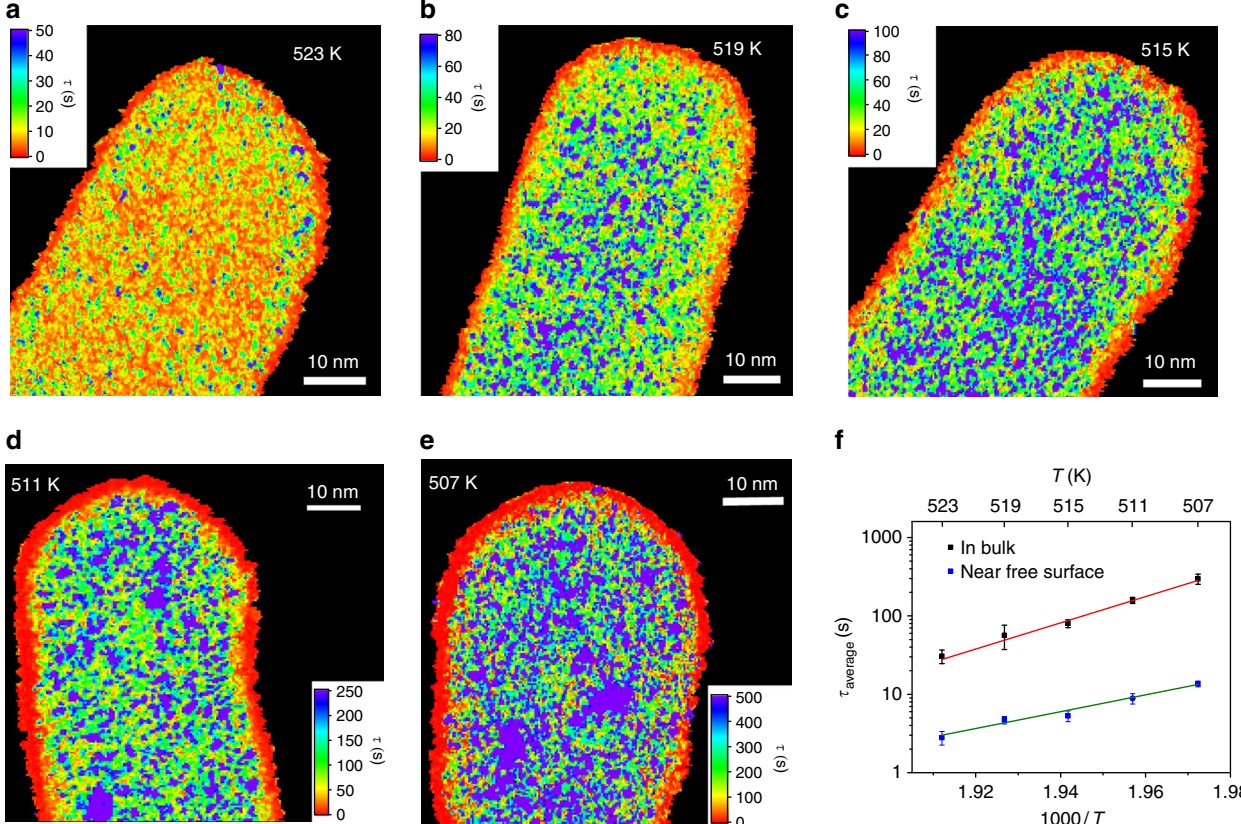

**Fig. 2** Spatial maps of structural relaxation time on the nanowire in the supercooled liquid regime. **a** $T = 523$ K, **b** $T = 519$ K, **c** $T = 515$ K, **d** $T = 511$ K, **e** $T = 507$ K. The maps show domains with varying relaxation time at the nanometer scale. With decreasing temperature, slow domains appear larger and occupy a greater fraction of the map, especially very close to $T_g = 507$ K. There is a region ~1 nm thick with ~20 times shorter relaxation time near the surface of every wire. **f** The mean structural relaxation time for the nanowire interior (bulk) and the near-surface layer. The error bars are the standard deviation of the mean of four measurements on different nanowires. Fitting to the Arrhenius form yields activation energies of 3.7 ± 0.3 eV for the bulk and 1.7 ± 0.3 eV for the near-surface

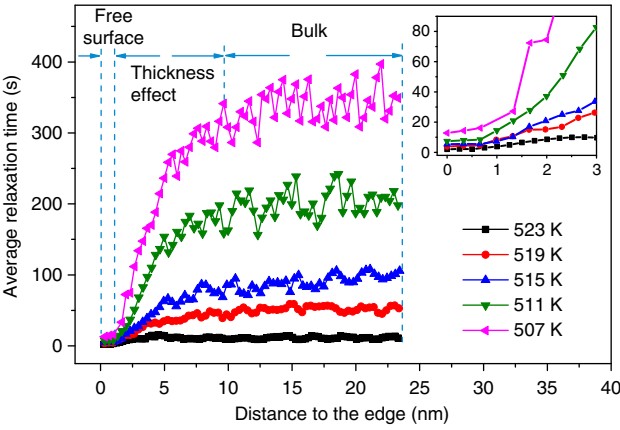

**Fig. 3** Near-surface region with fast dynamics. Profiles through the τ maps in Fig. 2. Near the free surface, the relaxation time remains almost constant and is about 20 times smaller than in the center of the nanowire. The inset shows that the near free surface layer is 3–4 pixels wide, which is between 0.7 and 1 nm. The dependence of τ on the projected thickness of the wire along the profile is discussed in the supplemental information

## Discussion

In the column approximation, each pixel in Fig. 2 arises from a cylinder of material with a diameter of 0.7 nm, extending through the thickness of the sample. However, not all of that material contributes to the intensity in the image at every time. Only a small portion of each cylinder contains atoms arranged in such a way that they scatter into the ~1 speckle that passes through the objective aperture and contributes to the image. Most of the atoms either do not have constructive interference of their scattering at all, or they scatter in a different direction into a different speckle that is blocked by the aperture. We estimate that the number of ordered clusters scattering through the aperture from one column is between 1 and 2 for the thickness of nanowires used here (Supplementary Note 4). Experimentally, the profiles of τ as a function of position in from the edge of the nanowire (and thus increasing thickness) in Fig. 3 and Supplementary Figs. 4 and 5, show that there is a systematic variation in τ for small thicknesses, but we can identify a region in the center of each nanowire where the thickness effects are negligible. Supplementary Fig. 6 shows that in this central region, the measured τ is independent of the sample thickness for thicknesses between 28 and 45 nm, including the 40 nm thick wires analyzed more extensively. Finally, Supplementary Fig. 5 is a τ image of a rectangular cross-section nanowire with uniform thickness, demonstrating that the observed fluctuations in τ are not caused by local thickness variations.

The average $\tau(T)$ derived from central portion of the images, shown in Fig. 2f, is in reasonable agreement with bulk measurements on the same alloy. τ extracted from the bulk increases from ~20 to ~350 s when the temperature is cooled from $T_g + 16$ K to $T_g$, which is consistent with data reported for supercooled metallic glass forming liquids[20,38–40]. $\tau(T)$ follows the Arrhenius law over the limited temperature range accessible in ECM, with an activation energy of $3.7 \pm 0.3$ eV, in good agreement with typical values for bulk metallic glass alloys[41,42], and the viscosity estimated from the Debye-Stokes Einstein equation is in reasonable agreement with estimates in the literature for the same alloy[43].

Quantitative characterization of spatially heterogeneous dynamics in simulations has been accomplished using high-order correlation functions[44–48]. We have calculated a similar two-time, two-position correlation function to derive a characteristic

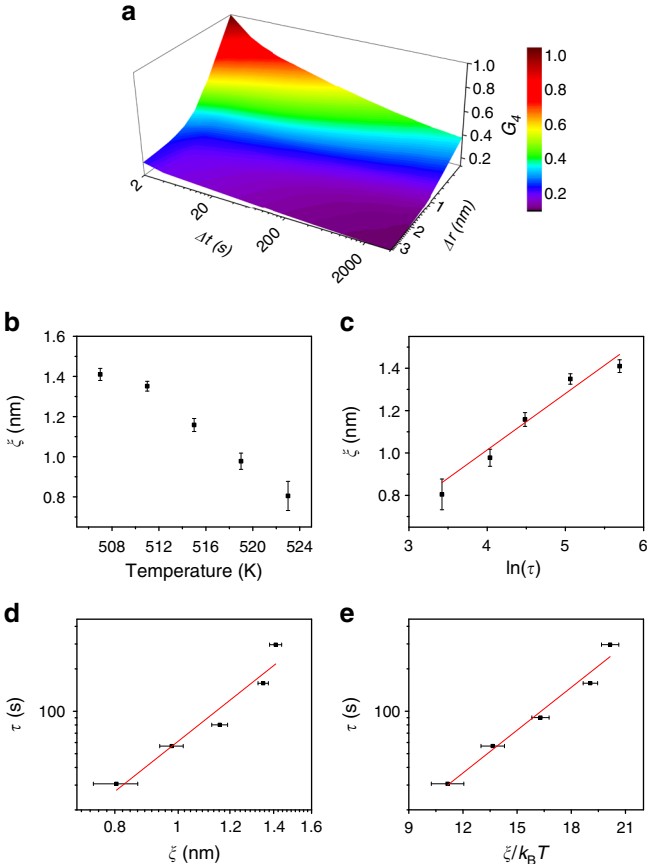

**Fig. 4** Characteristic length and time scale for heterogeneous dynamics. **a** An example $g_4(\Delta t, \Delta r)$ at 507 K. **b** The correlation length ξ as a function of temperature. **c** ξ vs. $\ln(\tau)$, which is a straight line in the Adam-Gibbs theory.[15] **d** A log–log plot of ξ vs. τ, which is a straight line in the inhomogeneous mode-coupling theory[54]. **e** A semi-log plot of τ vs. $\xi/k_B T$, which is a straight line in the random first-order transition theory[14]. The data are in reasonable agreement with all of the models. The reduced $\chi^2$ is 0.914, 0.905, and 0.961 for the fitting in **c**, **d**, **e**, respectively. The error bars are the standard deviation of the mean of four measurements on different nanowires

dynamic correlation length ξ

$$g_4(\Delta t, \Delta r) = \frac{\sum_{t,r} I(t,r)I(t+\Delta t,r)I(t,r+\Delta r)I(t+\Delta t,r+\Delta r)}{\sum_{t,r} I(t,r)\sum_{t,r} I(t+\Delta t,r)\sum_{t,r} I(t,r+\Delta r)\sum_{t,r} I(t+\Delta t,r+\Delta r)},$$

(3)

in which $I$ is the intensity of position $\vec{r}$ in the image, $t$ is the time of a frame in the diffraction time series, and $I(t, r+\Delta r)$ is the average intensity from all the pixels a distance $\Delta r$ from position $\vec{r}$. In simulation, functions like $g_4$ are calculated for the same particle (s) as a function of time, which we cannot do in experiments. However, ensemble averaging over many positions and long times converges to the same shape of function, as is well known for $g_2(t)$, the self-intermediate scattering function (in simulations)[49] and the intermediate scattering function (in experiments)[50].

Figure 4a shows an example $g_4(\Delta t, \Delta r)$ at 507 K. We extracted the spatial decay of $g_4$ by fixing the $\Delta t = \tau(T)$ and obtained the correlation length ξ through the exponential fitting as described in Supplementary Fig. 7 and Supplementary Note 5. We also extracted ξ and a characteristic time ν simultaneously from the two-dimensional shape of $g_4$ as described in Supplementary Fig. 8 and Supplementary Note 6. The results are very similar and the conclusions are the same (Supplementary Fig. 9).

$\xi(T)$ increases from ~0.82 nm at 523 K to ~1.41 nm at 507 K, as shown in Fig. 4b. This range of length scales and general behavior is broadly consistent with previous theoretical and experimental estimates[11,15,51]. However, $\xi$ at the temperature closest to $T_g$ does not increase as strongly as expected. Further experiments with better temperature stability, longer data series, and potentially other alloys are required to determine if this single point represents new physics or if it is some limitation of the current experiments. Various microscopic models of dynamics in the supercooled liquid and the glass transition predict different behavior for $\xi(\tau)$. Figure 4c shows the linear fit between $\ln(\tau)$ with $\xi$, predicted by the Adam-Gibbs theory of cooperatively rearranging regions[15]. The data follow the prediction well, except for the data nearest to $T_g$ (longest $\xi$ and $\tau$). The inhomogeneous mode-coupling theory (IMCT)[35] predicts a power-law relationship. Figure 4d shows a log–log plot of $\xi(\tau)$, which is also a good fit at high $T$ but deviates close to $T_g$. The random first-order transition (RFOT) theory[14] predicts $\tau \sim \exp(k(\xi/k_BT)^n)$. Figure 4e shows $\ln(\tau)$ as a function of $\xi/k_BT$. A line fits over the entire data range. The reduced $\chi^2$ for all three fits is similar (0.914 for Adam-Gibbs, 0.905 for IMCT and 0.961 for RFOT due to the additional fitting parameter), so we cannot draw firm conclusions on the correct model at this time. Future experiments over a wider range of $\tau$ and $T$ may provide further insight.

The other striking feature in Fig. 2 is a near-surface layer with much faster dynamics than present in the bulk. The width of the near-surface layer is ~1 nm, as shown in Fig. 3, and $\tau$ varies from a few seconds at high temperature to ~20 s near $T_g$. Surface diffusion in solids is often faster than bulk diffusion, both for crystals[52] and for glasses[36,53,54], and similar observations have been made in liquids[55]. However, the near-surface layer observed here involves ~1014 atoms, of which only ~200 are in the first atomic layer in contact with vacuum, so it is unlikely that surface diffusion dominates. Nor is the layer an effect of the shape of the wire, since a similar fast near-surface layer is observed on a rectangular wire (Supplementary Fig. 5). From Fig. 2f, the activation energy for the near-surface layer is 1.7 ± 0.3 eV, consistent with the prediction from the RFOT theory that the free-energy barrier for activated motion near a free surface should be half that of the bulk[56]. A similar highly mobile but significantly thicker surface layer has been proposed as an explanation for the suppression of $T_g$ in very thin polymer films[57].

It has been widely observed that crystallization typically proceeds inwards from the surface[58], a phenomenon that is usually attributed to heterogeneous nucleation on the sample's surface and quantified by a reduction in the activation volume $f(\theta) = [0,1]$[59]. Surface initiated crystallization has been observed on nanowires similar to those studied here, and it has been used to explain the strong dependence of the crystallization rate on the diameter of the nanowire[60]. The enhanced surface mobility shown in Figs. 2 and 3 provides an alternative explanation for surface crystallization. To estimate the effect that the enhanced surface mobility has on the nucleation rate, we compare the classical homogenous nucleation rate in the surface layer with enhanced mobility to the heterogeneous nucleation rate with a catalytic surface site. The classical homogenous nucleation rate can be estimated by $I_{surfacemobility} \propto A/\eta_{surface} N_{surfacelayer} \exp(-\Delta G^*/kT)$, in which $I_{surfacemobility}$ is the nucleation rate in the ~1 nm thick near-surface layer, $A$ is a constant, $\eta_{surface}$ is the viscosity in the surface layer, $N_{surfacelayer}$ is the number of atoms in the surface layer, and $\Delta G^*$ is the activation energy for the formation of a stable nuclei. The heterogeneous nucleation rate is $I_{Het} \propto A/\eta N_{surface} \exp(-\Delta G^* f(\theta)/kT)$, in which $N_{surface}$ is the number of surface atoms and $\eta$ is the bulk viscosity. The condition $I_{surfacemobility} = I_{Het}$ allows us to calculate the required catalytic heterogeneous influence, quantified by $f(\theta)$, to match homogeneous

nucleation with a low-viscosity surface layer. With the parameters and temperature considered here, the rates are equal at $f(\theta) = 0.5$. When we consider non-classical models, which have been shown to be more appropriate in describing nucleation of deeply undercooled melts[62], such as present here, $I_{surfacemobility}$ approaches $I_{Het}$ with $f(\theta) \sim 1$. This suggests that enhanced surface mobility can explain widely observed surface nucleation equally well as heterogeneous nucleation. As the main evidence for heterogonous nucleation is the observation that nucleation proceeds from the surface, the results presented here suggest that this may need to be reconsidered, and that more sophisticated experiments are required to reveal the origin of widely observed surface nucleation in metals.

In summary, we demonstrate direct measurement at the nanoscale of spatial heterogeneous dynamics in the supercooled liquid state of Pt-based alloy nanowires using dark-field ECM. The dynamics are characterized by a growing length and time scale as the liquid cools toward the glass transition. The nanowires also exhibit a near-surface layer ~1 nm thick with substantially faster dynamics than the bulk. The near-surface layer provides an effective mechanism for surface crystallization of liquids by homogeneous, as opposed to heterogeneous, nucleation.

## Methods

**Sample preparation**. $Pt_{57.5}Cu_{14.7}Ni_{5.3}P_{22.5}$ glassy nanowires with diameter of 40–45 nm were fabricated by the nanomoulding method described in detail elsewhere[61]. As-fabricated, the nanowires were attached to a substrate plate of the same bulk metallic glass. The metallic glass plate was rinsed with distilled water and isopropyl alcohol to minimize the residual salts and anodized aluminum oxide from fabrication. Then the plate was immersed in methanol and nanowires were released by sonication for 15–20 min. The methanol containing nanowires was dropped onto the TEM heating chip through a micropipette (1.5–1.8 μL). After evaporation of methanol, some nanowires were randomly attached on the $SiN_x$ membrane of chip window, which is ~90% electron transparent. This process was repeated several times based on the density of nanowires in methanol to ensure that enough nanowires were attached and isolated. Some contamination was introduced during sample preparation, probably from the methanol, so the sample was plasma cleaned at 20 psi Ar + $O_2$ mixture for 12–15 min before microscopy experiments.

**Dark-field electron correlation microscopy**. The wires were heated into the supercooled liquid state between the glass transition temperature $T_g = 507$ K up to 523 K, near the crystallization temperature[37] using a DENSsolutions SH30 single-tilt heating holder, which can provide temperature stability of 0.1 K and temperature accuracy of 2%. Nanowires start to crystallize from the free surface when temperature increased to 527 K. The sample was equilibrated at temperature before data collection. At temperatures of 515, 511, and 507 K, the sample was heated to the target temperature at a rate of 20 K min$^{-1}$, and then held for 30–60 min before image time series acquisition. At 523 and 519 K, in order to avoid crystallization, the sample was first heated to 508 K at rate of 20 K min$^{-1}$ and held for 30–60 min, then heated to target temperature at the same heating rate and held isothermally for 2 min before data collection. At all temperatures, the equilibration time before data acquisition was at least five times the measured structural relaxation time shown in Fig. 2.

ECM measurements were carried out using tilted dark-field TEM imaging. Experiments used the University of Wisconsin-Madison FEI Titan with probe aberration corrector at 200 kV, operated in TEM mode. An objective aperture of 10 μm in diameter or 2.83 mrad of half angle was inserted, giving rise to speckles in the image ~0.7 nm in diameter, calculated from the Rayleigh criterion and confirmed by imaging. The speckle size sets the spatial resolution of the ECM experiment. An Orius 2.6 x 4 k fast CCD with 1 ms readout time was used to record the time series of images. In all, 256 by 256 pixel images were acquired with the magnification adjusted to yield a typical pixel size of 0.25 nm, so each speckle covers ~3 pixels.

The interval between frames in the image series was set as a function of temperature to ~$0.005\tau_{med}$, where $\tau_{med}$ was acquired with a time interval short enough not to influence the results[30]. The acquisition time per frame was set to 0.1, 0.25, 0.5, 1, and 2 s for temperatures 523, 519, 515, 511, and 507 K respectively. Every image series consists of ~4000 frames. The total time for the time series was set to ~$20\tau_{med}$, to provide a balance between a time series that is too long, which may average together short and long relaxation process over temporally fluctuating dynamics, yielding artificial spatial homogeneity in the relaxation time[62], and a time series that is too short, which yields autocorrelation functions that are not well converged[10,63]. In all, $20\tau_{med}$ can be thousands of seconds near $T_g$, so rigid image

alignment was used to correct sample drift. Supplementary Movie 1 is a typical image series with 4000 frames acquired at 523 K. The bright spot in the image arises from a crystallized chunk of nanowire, the intensity of which does not change over the whole image series. In addition, because the background scattering intensity from $SiN_x$ is quite small compared with intensity from sample, the edge of nanowire can be well defined. Therefore, with the bright spot and the outline of nanowire as references, drift correction can be realized through rigid image alignment with single pixel precision using the convolution-based alignment in DigitalMicrograph software. Supplementary Movie 2 shows the series in Supplementary Movie 1 after alignment. The intensity of the pixels fluctuates, but the outline of the nanowire is stationary. Drift correction enables acquisition and analysis of long data series at low temperatures.

**Determining structural relaxation time.** $g_2(t)$ was calculated from the intensity time series at a single pixel $I(i)$ from

$$g_2(p) = \frac{(N-p)\sum_{i=0}^{N-p-1} I(i)I(i+p)}{[\sum_{i=0}^{N-p-1} I(i)][\sum_{i=0}^{N-p-1} I(i+p)]}, \quad (4)$$

where $N$ is the total number of frames in the diffraction time series, and $p$ and $i$ indicate position in the time series from 1 to $N$. $g_2(t)$ is related to intermediate scattering function $f(t)$ in an ergodic system by

$$g_2(t) = 1 + C[f(t)]^2, \quad (5)$$

where $C$ is an instrument-dependent parameter[64].

Structural relaxation kinetics and the intermediate scattering function in relaxation phenomena, which originate from a superposition of numerous and different relaxation processes such as in amorphous materials are generally described with the KWW function,

$$f(t) = f(0) \exp\left(-\frac{t}{\tau}\right)^\beta, \quad (6)$$

where $\tau$ is the structural relaxation time, and $\beta$ is the stretching exponent. Eq. (2) arises from substituting Eq. (6) into Eq. (5). To extract maps of $\tau$ and $\beta$, $g_2(t)$ calculated from Eq. (4) for every pixel, resampled logarithmically in time, then fit to Eq. (2) using standard non-linear least-squares fitting.

**Data availability**. All raw and analyzed data which support the findings of this study are available without restriction in the Materials Data Facility, DOI: 10.18126/M2GW5F[65].

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

# ARTICLE

43. Legg, B. A., Schroers, J. & Busch, R. Thermodynamics, kinetics, and crystallization of $Pt_{57.3}Cu_{14.6}Ni_{5.3}P_{22.8}$ bulk metallic glass. *Acta Mater.* **55**, 1109–1116 (2007).

44. Glotzer, S. C., Novikov, V. N. & Schrøder, T. B. Time-dependent, four-point density correlation function description of dynamical heterogeneity and decoupling in supercooled liquids. *J. Chem. Phys.* **112**, 509–512 (2000).

45. Keys, A. S., Abate, A. R., Glotzer, S. C. & Durian, D. J. Measurement of growing dynamical length scales and prediction of the jamming transition in a granular material. *Nat. Phys.* **3**, 260–264 (2007).

46. Lacevic, N., Starr, F. W., Schroder, T. B. & Glotzer, S. C. Spatially heterogeneous dynamics investigated via a time-dependent four-point density correlation function. *J. Chem. Phys.* **119**, 7372–7387 (2003).

47. Albert, S. et al. Fifth-order susceptibility unveils growth of thermodynamic amorphous order in glass-formers. *Sci. (80-.).* **352**, 1308–1311 (2016).

48. Toninelli, C., Wyart, M., Berthier, L., Biroli, G. & Bouchaud, J. P. Dynamical susceptibility of glass formers: Contrasting the predictions of theoretical scenarios. *Phys. Rev. E Stat. Nonlin. Soft Matter Phys.* **71**, 1–20 (2005).

49. Kob, W. & Andersen, H. C. Testing mode-coupling theory for a supercooled binary Lennard-Jones mixture. II. Intermediate scattering function and dynamic susceptibility. *Phys. Rev. E* **52**, 4134–4153 (1995).

50. Berthier, L. Direct experimental evidence of a growing length scale accompanying the glass transition. *Sci. (80-.).* **310**, 1797–1800 (2005).

51. Khouri, J. & Johari, G. P. Length scale of dynamic heterogemeties at the glass transition determined by multidimensional nuclear magnetic resonance. *J. Chem. Phys.* **138**, 2727–2730 (2013).

52. Mo, Y., Kleiner, J., Webb, M. & Lagally, M. Activation energy for surface diffusion of Si on Si (001): A scanning-tunneling-microscopy study. *Phys. Rev. Lett.* **66**, 1998–2002 (1991).

53. Zhang, W., Brian, C. W. & Yu, L. Fast surface diffusion of amorphous o-terphenyl and its competition with viscous flow in surface evolution. *J. Phys. Chem. B.* **119**, 5071–5078 (2015).

54. Zhang, Y. & Fakhraai, Z. Invariant fast diffusion on the surfaces of ultrastable and aged molecular glasses. *Phys. Rev. Lett.* **118**, 1–5 (2017).

55. Cooper, J. T. & Harris, J. M. Imaging fluorescence-correlation spectroscopy for measuring fast surface diffusion at liquid/solid interfaces. *Anal. Chem.* **86**, 7618–7626 (2014).

56. Stevenson, J. D. & Wolynes, P. G. On the surface of glasses. *J. Chem. Phys.* **129**, 234514 (2008).

57. Salez, T., Salez, J., Dalnoki-Veress, K., Raphaël, E. & Forrest, J. a. Cooperative strings and glassy interfaces. *Proc. Natl Acad. Sci.* **112**, 8227–8231 (2015).

58. Kelton, K. F. Crystal Nucleation in Liquids and Glasses. *Solid State Physics* **45**, 75–177 (1991).

59. Gránásy, L., Egry, I., Ratke, L., & Herlach, D. M. On the diffuse interface theory of nucleation. *Scr. Metall. Mater.* **30**, 621–626 (1994).

60. Sohn, S. et al. Nanoscale size effects in crystallization of metallic glass nanorods. *Nat. Commun.* **6**, 8157 (2015).

61. Assadi, H. & Schroers, J. Crystal nucleation in deeply undercooled melts of bulk metallic glass forming systems. *Acta Mater.* **50**, 89–100 (2002).

62. Kumar, G., Tang, H. X. & Schroers, J. Nanomoulding with amorphous metals. *Nature* **457**, 868–872 (2009).

63. MacKowiak, S. A. & Kaufman, L. J. When the heterogeneous appears homogeneous: Discrepant measures of heterogeneity in single-molecule observables. *J. Phys. Chem. Lett.* **2**, 438–442 (2011).

64. Lu, C. -Y. & Vanden Bout, D. A. Effect of finite trajectory length on the correlation function analysis of single molecule data. *J. Chem. Phys.* **125**, 124701 (2006).

65. Grübel, G. & Zontone, F. Correlation spectroscopy with coherent X-rays. *J. Alloy. Compd.* **362**, 3–11 (2004).

66. Zhang, P., Maldonis, J. J., Liu, Z., Schroers, J. & Voyles, P. M. Data in spatially heterogeneous dynamics in a metallic glass forming liquid imaged by electron correlation microscopy. (2018). Available at: https://doi.org/10.18126/M2GW5F

## Acknowledgements

We thank Mark Ediger for helpful discussions. This work was supported by the US National Science Foundation under Contract No. DMR-1506564 (P. Zhang and P. M. Voyles). The facilities and instrumentation for TEM sample preparation and microscopy at UW-Madison were supported by the University of Wisconsin Materials Research Science and Engineering Center (DMR-1121288). Preparation of metallic glass nano-wires by Ze Liu and Jan Schroers was supported by the Department of Energy through the Office of Basic Energy Sciences (#DE SC0004889).

## Author contributions

P.Z. and P.M.V. conceived the study. P.Z. performed the ECM experiments and data analysis. J.J.M. developed and implemented the four-body correlation function. Z.L. and J.S. synthesized the metallic glass nanowires. P.Z. and P.M.V. wrote the manuscript, which all authors revised and approved.

## Additional information

**Competing interests:** The authors declare no competing interests.

