## [Peer Review File(PDF 413 kb) · Nature Communications]

Reviewer #1 (Remarks to the Author):

The manuscript "Spatial Heterogeneous Dynamics in a Metallic Glass Forming Liquid Imaged by Electron Correlation Microscopy" by Pei Zhang et al reports on the direct experimental visualization of the spatial dynamical heterogeneity in a supercooled metallic liquid.

Dynamical heterogeneity is a hot topic in the physics of glass transition and the reported findings are of high relevance for the broad community. Moreover the obtained results seem to agree with the prediction of the RFOT theory.

From the experimental point of view the authors used a new technique - dark-field electron correlation microscopy which is well suited for the problem addressed. The results show clearly the spatial dynamical heterogeneity on the 0.95-1.5 nm . Yet I could not understand the origin of the observed heterogeneity. If the intensity in each pixel results from the projected full thickness of the nanowire (~40 nm) this includes ~40 dynamical clusters then I would expect the averaging over 40 clusters to smooth the spatial dynamical heterogeneity. How one can observe spatial heterogeneity on ~1 nm scale when averaging over 40 clusters? This is a critical question that requires clarification.

Below I list less critical comments and questions:

1) Line 39. The authors write "It is the electron equivalent of (x-ray) photon correlation spectroscopy ..." This is true for original EMC version described in ref.27 and ref.28 but in the presented work the authors used dark-field image based fluctuation electron microscopy ref.33 (Line 62). So it is confusing and rearranging the two paragraph between lines 38 and 65 to improve the clarity would be much appreciated.

2) It is not stated why authors chose Pt_{57.5}Cu_{14.7}Ni_{5.3}P_{22.5} system if "At higher temperatures, the nanowires crystallized before a long enough time series could be acquired". Justification of the system is needed for non-expert reader.

3) It would be very convincing if authors could show the two extreme cases: 1) a measurement of the dynamics at higher temperature where the spatial dynamical fluctuations are expected to disappear or dynamics is too fast to detect ($g_2(t)=1$) and 2) a measurement below T_g to see the frozen dynamics $g_2(t)=1+A$.

4) Line 129 $I(r+\Delta r)$ does not appear in the formula (3) The author should clarify how this intensity is related to the formula (3).

5) Figure 4(a) might look better if the axes Δt and Δr are in log scale.

6) From the current understanding of the glass transition it is expected that the length of the spatial heterogeneity grows upon approaching the T_g . But the data in Figure 4 (b) show deviation from the expected behavior at $T=507$ K. I would like the authors to comment on this result.

7) In the Supplementary materials line 204 the authors write "... β is between 0 and 1, so we do not observe $\beta > 1$ as reported in XPCS experiments on metallic glasses.ref 10." This claim is misleading because in the ref. 10 the $\beta > 1$ is observed in the glassy state which is not studied in the present work, while ref. 10 report $\beta < 1$ in the supercooled liquid state similar to the present work.

8) In Supplementary materials Figure S4 (b) shows a single frame dark field image. It is also informative to present a time average dark field image. I expect it to be rather smooth.

Reviewer #2 (Remarks to the Author):

This is a very interesting study concerning direct observation of dynamic heterogeneity in a metallic glass-former. The experimental technique is quite impressive, and the paper should be published.

I make suggestions that must be taken into account before I can definitively recommend the paper.

1) Literature: it seems to me that papers related to measurements of dynamic heterogeneity lengthscales in glass-formers are missing: the recent work on nonlinear susceptibilities (e.g. Science 352 (6291), 1308-1311 2016), linear response functions (e.g. Physical Review E 76 (4), 041510 234 2007), and the analog work using light scattering in colloidal materials by the Cipelletti group (there are several relevant papers of spatially/temporally resolved scattering techniques - which are maybe more relevant than numerical data).

2) The way the length is extracted from g_4 measurements using contour plots is not what people have done in the past, and is thus very confusing. One must instead fix the time Δt near $\tau_\alpha(T)$ at each T and fit the spatial decay of g_4 . As a refinement, one can also then vary Δt and deduce a time dependent length $\xi(\Delta t)$, see e.g. Phys. Rev. E 71, 041505 (2005).

3) The exponent n used in fig 4e is not predicted by theory (r_{fot}), and its determination in simulations totally meaningless (as they are performed in a regime where r_{fot} does not apply). Thus the strong claim that the present work is "best" described by r_{fot} should be toned down accordingly. This would not decrease the value of the work... quite the opposite.

All the revisions are marked in red in the manuscript, figure and captions, and supplementary materials.

Reviewers' comments:

Reviewer #1 (Remarks to the Author):

The manuscript "Spatial Heterogeneous Dynamics in a Metallic Glass Forming Liquid Imaged by Electron Correlation Microscopy" by Pei Zhang et al reports on the direct experimental visualization of the spatial dynamical heterogeneity in a supercooled metallic liquid. Dynamical heterogeneity is a hot topic in the physics of glass transition and the reported findings are of high relevance for the broad community. Moreover the obtained results seem to agree with the prediction of the RFOT theory.

From the experimental point of view the authors used a new technique - dark-field electron correlation microscopy which is well suited for the problem addressed. The results show clearly the spatial dynamical heterogeneity on the 0.95-1.5 nm. Yet I could not understand the origin of the observed heterogeneity. If the intensity in each pixel results from the projected full thickness of the nanowire (~40 nm) this includes ~40 dynamical clusters then I would expect the averaging over 40 clusters to smooth the spatial dynamical heterogeneity. How one can observe spatial heterogeneity on ~1 nm scale when averaging over 40 clusters? This is a critical question that requires clarification.

We attempted to answer this question in the original submission in the supplemental material on p. 4-7, but clearly the explanation was not sufficiently clear. We will expand on that explanation here, and we have revised the section of the manuscript discussing this topic (p. 5) in the hopes of increasing clarity.

The key concept is that the measurement selects clusters in both real space and reciprocal space. The real space selection is in the form of the image of the nanowire created by the microscope objective lens and the CCD camera. This image has lateral resolution of ~0.7 nm but projects through the thickness of the sample. As the reviewer points out, if we consider dynamical clusters ~1 nm in diameter and a sample thickness of 40 nm, then each pixel contains a contribution from up to 40 clusters.

The reciprocal space selection is created by the objective aperture shown in Figure 1(a). The aperture is superimposed on the diffraction pattern from the sample. It admits only electrons diffracted into a (fairly small) range of angles into the image. Based on a simple kinematic diffraction argument accounting for the radius of the aperture and the size of the clusters,¹ only ~5% of randomly-oriented, 1 nm diameter clusters will diffract through the aperture and contribute to the image. That predicts that 2 clusters at a time will contribute intensity to each pixel, not 40. Similar arguments led Hirata and Chen to similar conclusions regarding the fraction of clusters through the thickness of a TEM sample that contribution to nanobeam

electron diffraction experiments.² Separation in reciprocal space alone is regularly invoked in the context of coherent x-ray scattering of disordered, fluctuating systems like colloids (see *e.g.* Wochner *et al.*³).

The experimental results in Figure S4 show that the mean relaxation time is independent of the nanowire diameter over the range of 28-45 nm. Based on the estimate above, we expect that over this thickness range, the number of clusters contributing to each pixel is around 1. For larger thicknesses, the average relaxation time is suppressed because several clusters regularly contribute to each pixel. The nanowires we have studied are within the thickness range for which the relaxation time is thickness-independent. Moreover, the mean relaxation time is in agreement with previous measurements of the viscosity of the same alloy, as noted in the manuscript on p. 6. If we were not measuring the fundamental structural relaxation time due to cluster-overlap effects, the data would not agree with bulk measurements.

Below I list less critical comments and questions:

1) Line 39. The authors write "It is the electron equivalent of (x-ray) photon correlation spectroscopy ..." This is true for original EMC version described in ref.27 and ref.28 but in the presented work the authors used dark-field image based fluctuation electron microscopy ref.33 (Line 62). So it is confusing and rearranging the two paragraph between lines 38 and 65 to improve the clarity would be much appreciated.

We thank the reviewer for noting this inaccuracy in the manuscript. We have changed the description on p. 2 of the manuscript to read "ECM uses time-resolved coherent electron scattering to study liquid dynamics at the nanoscale. When performed in diffraction mode with a focused probe, it is the electron equivalent of (x-ray) photon correlation spectroscopy".

2) It is not stated why authors chose Pt57.5Cu14.7Ni5.3P22.5 system if "At higher temperatures, the nanowires crystallized before a long enough time series could be acquired". Justification of the system is needed for non-expert reader.

Crystallization always sets the high temperature limit for experiments on metallic glass supercooled liquids. Only superfast heating, potentially $\sim 10^6$ K/s, can avoid crystallization and drive the sample directly from the solid glassy state to the equilibrium high-temperature liquid,⁴ and high heating rates do not allow enough time to acquire an image time series. Any metallic glass alloy we selected would have this limitation. Other types of glasses, such as covalently-bonded silicates, do not have this behavior. They offer attractive prospects for future work, but those investigations lie outside the scope of the current manuscript.

Out of the metallic glasses, we chose $\text{Pt}_{57.5}\text{Cu}_{14.7}\text{Ni}_{5.3}\text{P}_{22.5}$ for gentle, ion-beam free TEM sample preparation, its relatively stability in the supercooled liquid state, and the data on its bulk properties available in the literature. $\text{Pt}_{57.5}\text{Cu}_{14.7}\text{Ni}_{5.3}\text{P}_{22.5}$ can be made into nanowires with different diameters, which are thin enough to be used as TEM samples without further thinning. We have also tried using focused ion beam milling to thin bulk $\text{Pt}_{57.5}\text{Cu}_{14.7}\text{Ni}_{5.3}\text{P}_{22.5}$ for ECM experiments, but the sample crystallized at lower temperatures than either the nanowires or the bulk alloy. We believe the suppressed crystallization is due to either surface damage or implanted Ga from the FIB. Revisions to this effect are in the manuscript on p. 3.

$\text{Pt}_{57.5}\text{Cu}_{14.7}\text{Ni}_{5.3}\text{P}_{22.5}$ is one of the most widely studied alloys for the thermoplastic forming method used to create the nanowires we studied. As a result, its viscosity and crystallization behavior have been widely studied, including in nanowire form.⁵⁻⁷ Simpler binary or ternary alloys are generally less stable against crystallization in the supercooled liquid state.

We plan to revisit bulk sample preparation for ECM experiments and extend the current results to other compositions in future work.

3) It would be very convincing if authors could show the two extreme cases: 1) a measurement of the dynamics at higher temperature where the spatial dynamical fluctuations are expected to disappear or dynamics is too fast to detect ($g_2(t)=1$) and 2) a measurement below T_g to see the frozen dynamics $g_2(t)=1+A$.

We agree that these would be exciting experiments, but we have previously explored the limitations of our experimental approach that prevent us from making the suggested measurements. At higher temperatures in the supercooled liquid, the sample crystallizes, ending the experiment, before we can reach high enough temperature to achieve spatially uniform dynamics. Dynamics in the equilibrium liquid are much too fast for the time resolution of our experiment. We have investigated the behavior of our experiments when the time resolution is insufficient.⁸ In brief, $g_2(t)$ fails to converge, loses the KWW shape, and the measurement becomes meaningless.

At the low temperature end below T_g , there are two problems. The first is that the dynamics do not completely stop. Instead, we enter the regime of very slow dynamics associated with aging. These dynamics have recently been studied by Ruta and co-workers using XPCS in a beautiful series of papers.⁹⁻¹¹ These results show that aging dynamics will influence of ECM experiments. Unfortunately, the time scales are too long for us to measure using current techniques. As shown in Zhang *et al.* (2016), the total length of the experimental time series must be many times longer than the characteristic relaxation time. Otherwise, $g_2(t)$ again fails to converge, loses the KWW shape, and the measurement becomes meaningless. Spatial drift of the sample and instrument currently limit the duration of our experiments to a time that is too short to perform the suggested experiments below T_g . Extending the duration of the experiments is an interesting future

direction, but it requires methodological innovations that are outside the scope of the current work.

4) Line 129 $I(r+\Delta r)$ does not appear in the formula (3) The author should clarify how this intensity is related to the formula (3).

We thank the reviewer for pointing out this oversight. We have replace $I(r+\Delta r)$ to $I(t, r+\Delta r)$, which is the average intensity from all the pixels a distance Δr from position \mathbf{r} in the image acquired at time t .

5) Figure 4(a) might look better if the axes Δt and Δr are in log scale.

We have updated Figure 4(a) with log scale in Δt and Δr in the revised manuscript as suggested. The revised figure is also shown below as Figure 1.

Figure 1. An example $g_4(\Delta t, \Delta r)$ at 507 K plotted with log scale.

6) From the current understanding of the glass transition it is expected that the length of the spatial heterogeneity grows upon approaching the T_g . But the data in Figure 4 (b) show deviation from the expected behavior at $T=507$ K. I would like the authors to comment on this result.

This is an excellent question to which we do not have a fully satisfactory answer at this time. It may be that this data point results from some unexpected behavior in the liquid very close to T_g . Unfortunately, it could also be an experimental artifact of some kind, perhaps related to an insufficiently long time series to capture the longest relaxation times near T_g or the effects of long-time temperature fluctuations of the sample very close to T_g . On the plus side, the data in the manuscript are reproducible in different experiments on different nanowires. On the negative

side, longer data series are precluded by current experimental limitations, as mentioned above. Adding additional data points at other temperatures close to T_g is also not currently possible due to the limitations in the absolute accuracy of the temperature of the MEMS heating holder. Higher accuracy MEMS chips are nearing commercialization and methodologies for longer experiments are in development, so we should have additional insight eventually. Future experiments on other metal alloys or other classes of materials altogether may result in data over a wider range of relaxation time and characteristic length.

We do not want to over-interpret a single data point, so we do not wish to speculate too extensively on its meaning in the manuscript. However, we have added a very brief additional discussion on p. 7.

7) In the Supplementary materials line 204 the authors write "... β is between 0 and 1 , so we do not observe $\beta > 1$ as reported in XPCS experiments on metallic glasses.ref 10." This claim is misleading because in the ref. 10 the $\beta > 1$ is observed in the glassy state which is not studied in the present work, while ref. 10 report $\beta < 1$ in the supercooled liquid state similar to the present work.

The reviewer is entirely correct and we regret the error. In the revised manuscript, the text “so we do not observe $\beta > 1$ as reported in XPCS experiments on metallic glasses.ref 10.” is deleted.

8) In Supplementary materials Figure S4 (b) shows a single frame dark field image. It is also informative to present a time average dark field image. I expect it to be rather smooth.

As the referee anticipated, the average image over the time series is smooth. It is shown below compared to a single snapshot and included in supplemental Figure S6 in the revised version. The remaining contrast arises from small thickness variations in the wire, which normalize out of the correlation functions. We would add a time average dark field image as Figure S6(c) in the revision. The figure is also attached here and it looks smoother. Thank you for pointing this out.

Figure 2. The time-averaged dark field image taken from the ECM data series for a rectangular nanowire at $T = 519$ K.

Reviewer #2 (Remarks to the Author):

This is a very interesting study concerning direct observation of dynamic heterogeneity in a metallic glass-former. The experimental technique is quite impressive, and the paper should be published.

I make suggestions that must be taken into account before I can definitively recommend the paper.

1) Literature: it seems to me that papers related to measurements of dynamic heterogeneity lengthscales in glass-formers are missing: the recent work on nonlinear susceptibilities (e.g. Science 352 (6291), 1308-1311 2016), linear response functions (e.g. Physical Review E 76 (4), 041510 234 2007), and the analog work using light scattering in colloidal materials by the Cipelletti group (there are several relevant papers of spatially/temporally resolved scattering techniques - which are maybe more relevant than numerical data).

We appreciate the pointers to the literature provided by the referee. They are now included as references 26, 27, 49 and 50 in the revised manuscript and cited in the appropriate places.

2) The way the length is extracted from g_4 measurements using contour plots is not what people have done in the past, and is thus very confusing. One must instead fix the time Δt near $\tau_{\alpha}(T)$ at each T and fit the spatial decay of g_4 . As a refinement, one can also then vary Δt and deduce a time dependent length $\xi(\Delta t)$, see e.g. Phys. Rev. E 71, 041505 (2005).

We appreciate the reviewer's suggestion on this data interpretation. Our original approach was motivated by the fact that the exact form of g_4 we use does not reduce to $g_2(t)$ at $\Delta r = 0$. However, the referee's suggested method is definitely easier to explain, so we have implemented it and included it in the main text in Figure 4 and the accompanying discussion.

We extended the calculation of the four point correlation function g_4 to $\Delta t \sim 13\tau_{\text{med}}$ for each temperature. We extracted $g_4(\Delta r)$ at fixed Δt equal to the mean of relaxation time τ derived from g_2 , and then fit the spatial decay of g_4 as before to extract the characteristic length ξ . Figure 3 below and the revised Figure 4 in the manuscript show the new calculation of $\xi(T)$ and the fits to

the Adam-Gibbs, IMCT, and RFOT models. The results are numerically similar to the previous method and the functional dependence on temperature is indistinguishable. This result also demonstrates that varying Δt over the accessible range in our experiments creates very little change in ξ . The fit to all three models is similar. The previous analysis for the characteristic length and time has been moved to the supplemental materials (Figure S2 and accompanying discussion).

Figure 3. (a) An example $g_4(\Delta t, \Delta r)$ at 507 K. (b) The correlation length ξ as a function of temperature. (c) ξ vs. $\ln(\tau)$, fitting to the Adam-Gibbs theory. (d) A log-log plot of ξ vs τ , fitting to the inhomogeneous mode coupling theory (IMCT). (e) A semi-log plot of τ vs $\xi/k_B T$, fitting to the random first-order transition theory (RFOT). The data are in reasonable agreement with all of the models.

3) The exponent n used in fig 4e is not predicted by theory (rfot), and its determination in simulations totally meaningless (as they are performed in a regime where rfot does not apply). Thus the strong claim that the present work is "best" described by rfot should be toned down accordingly. This would not decrease the value of the work... quite the opposite.

We appreciate the reviewer's perspective on the agreement of our work with the models we have tested for the supercooled liquid. We have made the change in emphasis suggested both in the abstract and in the discussion of Figure 4 on p. 7. We are planning future work which may provide a more definitive test of these models, but it is far from complete!

References

1. Stratton, W. G. & Voyles, P. M. A phenomenological model of fluctuation electron microscopy for a nanocrystal/amorphous composite. *Ultramicroscopy* **108**, 727–736 (2008).
2. Hirata, A. & Chen, M. W. Angstrom-beam electron diffraction of amorphous materials. *J. Non. Cryst. Solids* **383**, 52–58 (2014).
3. Wochner, P. *et al.* X-ray cross correlation analysis uncovers hidden local symmetries in disordered matter. *Proc. Natl. Acad. Sci. U. S. A.* **106**, 11511–11514 (2009).
4. Johnson, W. L. *et al.* Beating crystallization in glass-forming metals by millisecond heating and processing. *Science* **332**, 828–33 (2011).
5. Shao, Z. *et al.* Size-dependent viscosity in the super-cooled liquid state of a bulk metallic glass. *Appl. Phys. Lett.* **102**, 221901 (2013).
6. Sohn, S. *et al.* Nanoscale size effects in crystallization of metallic glass nanorods. *Nat. Commun.* **6**, 9157 (2015).
7. Legg, B. A., Schroers, J. & Busch, R. Thermodynamics, kinetics, and crystallization of Pt_{57.3}Cu_{14.6}Ni_{5.3}P_{22.8} bulk metallic glass. *Acta Mater.* **55**, 1109–1116 (2007).
8. Zhang, P. *et al.* Applications and limitations of electron correlation microscopy to study relaxation dynamics in supercooled liquids. *Ultramicroscopy* **178**, 125–130 (2017).
9. Ruta, B. *et al.* Atomic-Scale Relaxation Dynamics and Aging in a Metallic Glass Probed by X-Ray Photon Correlation Spectroscopy. *Phys. Rev. Lett.* **109**, 165701 (2012).
10. Giordano, V. M. & Ruta, B. Unveiling the structural arrangements responsible for the atomic dynamics in metallic glasses during physical aging. *Nat. Commun.* **7**, 1–8 (2015).
11. Evenson, Z. *et al.* X-Ray Photon Correlation Spectroscopy Reveals Intermittent Aging Dynamics in a Metallic Glass. *Phys. Rev. Lett.* **115**, 175701 (2015).

Reviewer #1 (Remarks to the Author):

The authors addressed all my questions and the manuscript can be published.

Reviewer #2 (Remarks to the Author):

I think the authors have responded appropriately to the initial reports.

I recommend publication.